# The Potential of Caffeic Acid Lipid Nanoparticulate Systems for Skin Application: In Vitro Assays to Assess Delivery and Antioxidant Effect

**DOI:** 10.3390/nano11010171

**Published:** 2021-01-12

**Authors:** Supandeep Singh Hallan, Maddalena Sguizzato, Markus Drechsler, Paolo Mariani, Leda Montesi, Rita Cortesi, Sebastian Björklund, Tautgirdas Ruzgas, Elisabetta Esposito

**Affiliations:** 1Department of Chemical and Pharmaceutical Sciences, University of Ferrara, I-44121 Ferrara, Italy; hllsnd@unife.it (S.S.H.); maddalena.sguizzato@unife.it (M.S.); 2Biofilms—Research Center for Biointerfaces, Faculty of Health and Society, Malmö University, SE-20506 Malmö, Sweden; sebastian.bjorklund@mau.se; 3Bavarian Polymerinstitute “Electron and Optical Microscopy”, University of Bayreuth, D-95440 Bayreuth, Germany; markus.drechsler@uni-bayreuth.de; 4Department of Life and Environmental Sciences, Polytechnic University of Marche, I-60131 Ancona, Italy; p.mariani@staff.univpm.it; 5Department of Life Sciences and Biotechnology, University of Ferrara, I-44121 Ferrara, Italy; leda.montesi@unife.it

**Keywords:** solid lipid nanoparticles, caffeic acid, ethosomes, Franz cell, skin, antioxidative reaction

## Abstract

The object of this study is a comparison between solid lipid nanoparticles and ethosomes for caffeic acid delivery through the skin. Caffeic acid is a potent antioxidant molecule whose cutaneous administration is hampered by its low solubility and scarce stability. In order to improve its therapeutic potential, caffeic acid has been encapsulated within solid lipid nanoparticles and ethosomes. The effect of lipid matrix has been evaluated on the morphology and size distribution of solid lipid nanoparticles and ethosomes loaded with caffeic acid. Particularly, morphology has been investigated by cryogenic transmission electron microscopy and small angle X-ray scattering, while mean diameters have been evaluated by photon correlation spectroscopy. The antioxidant power has been evaluated by the 2,2-diphenyl-1-picrylhydrazyl methodology. The influence of the type of nanoparticulate system on caffeic acid diffusion has been evaluated by Franz cells associated to the nylon membrane, while to evaluate caffeic acid permeation through the skin, an amperometric study has been conducted, which was based on a porcine skin-covered oxygen electrode. This apparatus allows measuring the O_2_ concentration changes in the membrane induced by polyphenols and H_2_O_2_ reaction in the skin. The antioxidative reactions in the skin induced by caffeic acid administered by solid lipid nanoparticles or ethosomes have been evaluated. Franz cell results indicated that caffeic acid diffusion from ethosomes was 18-fold slower with respect to solid lipid nanoparticles. The amperometric method evidenced the transdermal delivery effect of ethosome, indicating an intense antioxidant activity of caffeic acid and a very low response in the case of SLN. Finally, an irritation patch test conducted on 20 human volunteers demonstrated that both ethosomes and solid lipid nanoparticles can be safely applied on the skin.

## 1. Introduction

Phenolic compounds deriving from vegetable sources possess remarkable antioxidant properties, offering potential for many medicinal applications [1,2,3]. Among them, caffeic acid (CA), naturally found in coffee, fruits, plants, oils, grapes, and tea, is a hydroxycinnamic acid characterized by strong antioxidant power, due to radical scavenging activity and the inhibition of lipid peroxidation [4,5,6]. Thanks to this potential, CA can be employed in the treatment of UVB-induced photoaging and in the chemoprevention of malignant melanoma [7,8,9]. However, the poor water solubility and chemical stability of CA makes necessary the use of specialized delivery systems suitable for its solubilization, protection from degradation, and the maintenance of antioxidative power [10,11,12,13]. Recently, researchers have proposed nanoparticulate systems for antioxidant molecules encapsulation in order to preserve them from degradation and ensure a prolonged action [14,15]. Among the others, nanosystems based on natural lipids have gained particular attention due to their biocompatibility, capability to encapsulate lipophilic molecules, and easy way of production, avoiding the use of organic solvents, thus ensuring a low toxicity profile [16,17]. Additionally, lipid-based nanosystems are suitable for cutaneous application, due to the affinity of their matrix with the *stratum corneum* intercellular lipids [16]. Upon dispersion in water, lipids give raise to a variety of supramolecular structures organized in different crystalline phases [18]. Among the various colloidal systems that can be obtained depending on the composition and production method, solid lipid nanoparticles (SLN) and ethosomes (ETHO) deserve particular interest. Concerning SLN, their matrix is usually constituted of solid lipids, such as glycerides, sterols, fatty acids, or waxes [19,20,21,22]. Notably, SLN have been recently proposed for the oral and cutaneous administration of phenolic compounds, protecting them against chemical degradation [15,23]. On the other hand, ETHO are thermodynamically stable vesicular systems constituted of phospholipids, ethanol, and water [24]. The ETHO vesicles are organized as lipid bilayers connecting two hydrophilic regions, typified by an inner core and a dispersing phase, offering the possibility to solubilize molecules with different physical–chemical properties [24,25]. The ethanol presence (30–45% *v*/*v*) confers to ETHO vesicles a soft consistency, enabling their interaction with physiological membranes. Due to their potential, ETHO have been investigated as cutaneous delivery systems of many drugs, including phenolic compounds [11,26,27]. The cutaneous application of lipid nanosystems gives the opportunity to treat many dermatological disorders and pathologies. Indeed, the close interaction between the lipid matrix and the *stratum corneum* lipids promotes the passage of the active molecules through the skin [28]. In this regard, some authors have reported that the SLN matrix applied on the skin can fuse, thus forming a lipid film that exerts an occlusive effect, enabling the permeation of the encapsulated drugs [23,28]. Others suggest the formation of a drug reservoir in skin, depositing the drug in the *stratum corneum* [29]. Conversely, many studies have evidenced the transdermal effect of ETHO, allowing the deep penetration of drugs through the skin strata [25,30,31]. In line with this, ethanol is supposed to disorganize the *stratum corneum* barrier, opening spaces through which the soft and malleable ETHO vesicles can pass, as indicated by the presence of intact ETHO in the epidermis [25,32]. Different nanosystems have been investigated as carriers for phenolic compounds; however, to the best of our knowledge, the influence of the nanosystem type on the antioxidant behavior and transdermal potential of the phenolic compound has not been yet investigated. On this matter, the present study describes a comparison between SLN and ETHO for the cutaneous administration of CA. The effect of CA encapsulation has been evaluated, comparing size, morphology, and encapsulation efficiency. Furthermore, the physical–chemical stability of the two nano-systems as well as in vitro CA diffusion kinetics have been studied. Noteworthily, in order to compare the in vivo antioxidant potential of CA encapsulated in SLN and in ETHO, an amperometry-based approach has been employed. This recently developed electrochemical technique, based on the use of a pig skin covered oxygen electrode (SCOE), allows predicting the kinetics of antioxidant molecules penetration across the *stratum corneum* and its further engagement in anti-oxidative reactions, occurring in the skin in the presence of hydrogen peroxide (H_2_O_2_) [33,34]. The SCOE method enables measuring the alterations in O_2_ concentration within the skin due to the reactions of H_2_O_2_ with antioxidants, mimicking in this way their cutaneous administration [33]. In this study, the SCOE approach was employed to compare the penetration enhancer properties of SLN and ETHO upon cutaneous application, as well as the antioxidant power exerted by CA encapsulated in the two nanosystems. At last, in order to assess the safety of SLN and ETHO as cutaneous vehicles for CA, an in vivo irritation test was conducted.

## 2. Materials and Methods 

### 2.1. Materials

Caffeic acid (CA), glyceryl tristearate (tristearin), sodium citrate dihydrate, citric acid monohydrate, sodium chloride, phosphotungstic acid, catalase from bovine liver (H_2_O_2_:H_2_O_2_ oxidoreductase, mol wt 250 kDa) (CAT), and peroxidase from horseradish (mol wt 44 kDa) (PER) were purchased from Sigma-Aldrich (St Louis, MO, USA). The soybean lecithin (90% phosphatidylcholine) (PC) used for ETHO preparation was Epikuron 200 from Lucas Meyer (Hamburg, Germany). Pluronic F68 (PEO80-POP27-PEO80) (poloxamer) was obtained from BASF (Ludwigshafen, Germany). Nylon membranes (pore size 200 nm) and dialysis tubing cellulose membrane (size-100 feet, mol wt 12 kDa) were purchased from Millipore Sigma (St Louis, MO, USA). The Clark-type oxygen electrode was purchased from UAB “OPTRONIKA”, (Vilnius, Lithuania). All solutions were prepared in water purified by the Milli-Q system (Merck Millipore, Billerica, MA, USA) with a resistivity of 18.2 Ω cm. Solvents were of high-performance liquid chromatography (HPLC) grade, and all other chemicals were of analytical grade. 

### 2.2. Nanoparticulate Systems Preparation

Solid lipid nanoparticles (SLN) preparation was based on lipid melting, emulsification by homogenization, and sonication [10]. The emulsion was constituted of poloxamer (2.5% *w*/*w*) solution as the dispersing phase and tristearin (5% *w*/*w*, with respect to the whole weight of the dispersion) as the disperse phase. Namely, a poloxamer aqueous phase (4.75 mL) heated at 80 °C has been added to the molten lipid phase (250 mg) and mixed at 15,000 rpm at 80 °C for 1 min (IKA T25 digital ultraturrax, IKA-Werke GmbH & Co. KG, Staufen, Germany). The emulsion has been further homogenized by ultrasound at 6.75 kHz for 15 min (Microson ultrasonic Cell Disruptor-XL Minisonix, Bioz Inc., San Francisco, CA, USA) and stored at 25 °C. CA containing SLN (SLN-CA) was obtained by pouring and mixing CA into the molten lipid phase before the emulsification step. 

To prepare ethosome (ETHO), firstly, PC was solubilized in ethanol (3% *w*/*v*); afterwards, bidistilled water was added dropwise to the ethanol phase up to a final 70:30 (*v*/*v*) ratio [11]. The dispersion has been magnetically stirred at 750 rpm (IKA RCT basic, IKA^®^-Werke GmbH & Co. KG, Staufen, Germany) for 30 min at room temperature. CA containing ETHO (ETHO-CA) was obtained by adding the drug into PC ethanol solution before the dropwise addition of water. Both in SLN and in ETHO, the CA concentration was 0.1% *w*/*w*. 

### 2.3. Photon Correlation Spectroscopy (PCS) and Zeta Potential

Size distribution analysis of the nanoparticulate system has been obtained by a Zetasizer Nano S90 (Malvern Instruments, Malvern, UK) supplied with a 5-mW helium neon laser, wavelength output 633 nm. The measurements were repeated three times at 25 °C at a 90° angle; the “cumulant” method has been applied to decode the data [35]. Zeta potential values were obtained measuring the electrophoretic mobility according to the Hemholtz–Smoluchowski equation [36].

### 2.4. Electron Microscopy Analyses 

#### 2.4.1. Transmission Electron Microscopy (TEM)

For TEM analyses, SLN-CA and ETHO-CA were negatively stained by depositing a drop of sample on a TEM screen covered with a Formvar film (Media System Lab S.r.l. Macherio, MB, Italy). After 1 min, the excess drop was removed from the screen with filter paper in order to keep a very light veil of sample on the supporting substrate. A drop of 2% phosphotungstic acid was placed on the screen for 1 min and then removed with filter paper to surround the particles of the preparation deposited on the screen and adhere to their surface. The screen prepared in this way was observed with a ZEISS EM 910 transmission electron microscope (Carl Zeiss Microscopy, GmbH, Munich, Germany).

#### 2.4.2. Cryogenic Transmission Electron Microscopy (Cryo-TEM)

For Cryo-TEM analyses, SLN and ETHO samples have been firstly vitrified following a method previously described [10,11]. The vitrified species have been placed into a Zeiss EM922 Omega transmission electron microscope (Carl Zeiss Microscopy, GmbH, Munich, Germany) for imaging using a cryoholder (CT3500, Gatan, Pleasanton, CA, USA). Samples were kept at temperatures below −175 °C during the examination. The specimens have been evaluated with 1000–2000 e/nm^2^ doses at 200 kV. A charge-coupled device camera (Ultrascan 1000, Gatan) and an image processing system (GMS 1.9 software, Gatan) have been employed to digitally take the images. 

### 2.5. X-ray Scattering 

Small angle X-ray scattering (SAXS) analyses of SLN and ETHO have been performed at the bioSAXS beamline B21, at Diamond Light Source (Harwell, UK). CA loaded and unloaded SLN and ETHO were put into 0.5 ml tubes in an automated sample changer. Then, the samples were moved into a temperature-controlled quartz capillary and exposed for 1 s, acquiring 30 frames at 20 °C. Data were collected by a Dectris Eiger 4M (75 × 75 pixels) detector with a 3.7 m sample–detector distance and X-ray wavelength λ = 0.1 nm. The explored Q-range (Q is the modulus of the scattering vector, which is defined as 4π sin θ/λ, where 2θ is the scattering angle) extended between 0.026 and 4.6 nm^−1^. Two-dimensional data were corrected for background, detector efficiency, and sample transmission; afterwards, data were radially averaged to derive I(Q) vs. Q curves.

### 2.6. Encapsulation Efficiency and Loading Capacity of CA in SLN and ETHO

The encapsulation efficiency (EE) and loading capacity (LC) of CA in SLN and ETHO was determined by ultracentrifugation and HPLC [10,11]. Briefly, 500 μL aliquots of SLN-CA or ETHO-CA were poured into a centrifugal filter (Microcon centrifugal filter unit YM-10 membrane, NMWCO 3 kDa, Sigma-Aldrich, St. Louis, MO, USA) and centrifuged (Spectrafuge™ 24D Digital Microcentrifuge, Woodbridge, NJ, USA) for 20 min at 8000 rpm. Then, the lipid phase in the upper section of the filter was diluted in a 1:10 *v*/*v* ratio with methanol, in the case of SLN-CA, or ethanol, in the case of ETHO-CA, and kept under stirring for 2 or 0.5 h, respectively. After sample filtration by nylon syringe filters (0.22 μm pores), the amount of CA has been analyzed by high-performance liquid chromatography (HPLC), as below reported. The EE was determined as follows: EE = CA/T_CA_ × 100(1)
where CA is the amount of drug retained in SLN-CA or in ETHO-CA and T_CA_ is the total content of CA.

The LC was determined as follows:LC = CA/LP × 100(2)
where CA is the amount of drug measured by HPLC and LP is the amount of lipid phase in SLN-CA or in ETHO-CA.

### 2.7. Stability Studies

The physical–chemical stability of nanoparticulate systems stored at 22 °C for 6 months was investigated. Particularly size stability was determined by PCS periodically, evaluating the Z average diameter and dispersity index of SLN, SLN-CA, ETHO, and ETHO-CA. In addition, zeta potential has been measured 6 months after nanoparticulate systems production. To evaluate CA chemical stability, its content was quantified by the disaggregation of SLN-CA with methanol (1:10 *v*/*v*) or dilution of ETHO-CA with ethanol (1:10 *v*/*v*) and kept under stirring for 2 or 0.5 h, respectively. A CA aqueous solution (0.5 mg/mL) (CA-water) was taken as control. After sample filtration by nylon syringe filters (0.22 μm pores), the amount of CA has been analyzed by HPLC, as reported below. 

### 2.8. In Vitro Diffusion Experiments

CA diffusion from SLN-CA and ETHO-CA was studied using Franz diffusion cells (orifice diameter 0.9 cm; PermeGear Inc. Hellertown, PA, USA) associated to nylon membranes (pore diameter 0.2 μm). The membrane, previously hydrated for 1 h, was mounted on the lower receiving section of the cell previously filled with 5 mL of bidistilled water. This solution was stirred at 500 rpm by a magnetic bar and thermostated at 32 ± 1 °C [37]. After mounting the upper donor section on the lower by means of a clamp, 1 mL of each formulation (CA-water, SLN-CA, and ETHO-CA) was poured on the membrane surface. Then, the donor compartment was sealed to avoid evaporation. Two hundred microliters samples of receiving phase were withdrawn at predetermined time intervals (1–8 h) and analyzed for CA content using HPLC. Each removed sample was replaced with an equal volume of fresh bidistilled water. The CA concentrations were determined six times in independent experiments, and the mean values ± standard deviations were calculated. Then, the mean values were plotted as a function of time. Flux values were computed from the slope of the linear portion of the accumulation curves, while diffusion coefficients were calculated dividing fluxes by the CA concentration (mg/mL) in the analyzed form. 

### 2.9. HPLC Procedure

For HPLC analyses, a two-plungers alternative pump (Agilent Technologies 1200 series, USA), a UV-detector operating at 325 nm, and a 7125 Rheodyne injection valve with a 50 μL loop were employed. A stainless-steel C-18 reverse-phase column (15 cm length × 0.46 cm diameter) packed with 5 μm particles (Platinum C18, Apex Scientific, Alltech, Ronkonkoma, NY, USA) was eluted with a mobile phase containing acetonitrile/water 20:80 *v*/*v*, pH 2.5 at a flow rate of 0.7 mL/min. Retention time of CA was 4.5 min.

### 2.10. Antioxidant Activity

The antioxidant activity of CA encapsulated in SLN or in ETHO was evaluated considering its scavenging capacity toward the radical α,α-diphenyl-b-picrylhydrazyl (DPPH•), following the method of Marinova and Batchvarov with some modifications [38]. DPPH• was dissolved in ethanol (0.7 mg/mL) to obtain a stock solution. The stock solution was diluted with ethanol (1:10, *v*/*v*) until obtaining a 1.0-unit absorbance at 517 nm. SLN-CA and ETHO-CA were diluted with ethanol and mixed at 400 rpm with the same amount of DPPH solution (25 µL) for 30 min at 21 °C in the dark. The decreasing of absorbance was measured at 517 nm, using ethanol as the blank control. All assays were conducted measuring the absorbance thrice in a microplate reader. The radical scavenging activity was expressed as percentage inhibition of DPPH absorbance:DPPH scavenging activity = (A control − A sample)/A control × 100%(3)
where A control refers to the absorbance of the control (DPPH solution), while A sample stands for the absorbance of the sample (DPPH solution plus SLN-CA or ETHO-CA). The ability to scavenge the DPPH• was evaluated using different sample concentrations. Particularly, the IC_50_ values (i.e., the sample concentration required to scavenge 50% of the DPPH-free radicals) were obtained, expressing values as ascorbic acid equivalents, by linear regression analysis. The absorbance values have been measured with a Safire plate reader (Tecan Trading AG, Männedorf, Switzerland), and the IC_50_ for ascorbic acid was roughly 6.1 μg/mL [39]. 

### 2.11. Skin Resistance Measurements

Skin resistance measurements have been conducted using electrochemical impedance spectroscopy (EIS). The measurements have been made with the skin membrane mounted in Franz diffusion cell (PermeGear Inc. Hellertown, PA, USA) equipped with 4 electrodes (Appendix A) [40]. Briefly, impedance measurements were carried out using a potentiostat from Ivium Technologies (Eindhoven, Netherlands) within a frequency range 0.1 Hz–1 M Hz and 10 mV voltage amplitude. The temperature 32 ± 1 °C has been maintained throughout the measurements. The interactions of SLN and ETHO with the skin membrane were evaluated measuring skin membrane resistance. The donor chamber was filled with citrate buffer saline (CBS, 10 mM sodium citrate dihydrate, citric acid monohydrate, and 150 mM sodium chloride, pH 5.5) (*stratum corneum* site of skin), while phosphate buffer saline (PBS, sodium chloride–disodium hydrogen phosphate–potassium chloride–potassium dihydrogen phosphate (1:1:1:1), pH 7.4) was poured in the receptor chamber (dermal part of skin) to mimic physiological conditions. The impedance spectra of the skin membrane were recorded for 30 min (5 min interval); afterwards, CBS was replaced by SLN or ETHO. Then, the impedance measurements were carried on for 70 min in the presence of the nanosystems. Lastly, SLN or ETHO were again replaced by CBS, followed by the recording of impedance data for 20 min. Alterations in skin and solution resistance were evaluated using the data analysis program included in Ivium Potentiostat software. Specifically, skin resistance was determined by fitting impedance vs. frequency data (Appendix A) to an equivalent circuit considering the solution resistance, (R_sol_), connected in series with skin membrane impedance (Appendix A). The skin membrane impedance was represented as a parallel combination of a resistor (for skin membrane resistor, R_mem_) and a constant phase element (CPE). The change in skin membrane resistance (Ohm) was plotted vs. time (min). 

### 2.12. In Vitro Assessment of CA Antioxidant Activity in Skin Using SCOE

A SCOE apparatus was employed as previously described with some modifications [33,34,41]. Briefly, an oxygen electrode was covered with excised porcine skin membrane (500 µm thickness, 16 mm diameter) and fixed by a rubber o-ring. Amperometric measurements were performed under immersion of the assembled SCOE into an electrochemical measurement glass cell containing 10 mL of CBS (10 mM sodium citrate dihydrate, citric acid monohydrate, and 150 mM sodium chloride, pH 5.5) under constant agitation with magnetic stirrer. The current of the electrode was recorded using an AMEL model 2059 potentiostat/galvanostat (AMEL, Milano, Italy) by applying −0.7 V (vs. Ag/AgCl) on the Pt electrode of the oxygen electrode. After achieving the stable baseline, the response to 0.5 mM H_2_O_2_ was recorded, and CA (0.5 mM) in ethanol:CBS (30/70 *v*/*v*) solution (Sol-CA), SLN, SLN-CA, ETHO, and ETHO-CA were measured. All measurements were conducted at 32 °C. From the current response of the SCOE (see Appendix A for representative data), the lag time, t*_lag_* (i.e., the period required to establish a linear concentration gradient across the membrane) was calculated (Appendix A). The time zero was assigned to the moment of CA addition into the measurement cell, and the delta current values were integrated over the experimental time. The integral vs. time curve became linearly dependent on the time, when approaching a steady state of the current response. This linear part of the integral vs. time dependence was fitted to a linear equation and extrapolated toward a time-axis (Appendix A). The intercept was taken as a t*_lag_* value. The D*_app_* was calculated according to the following equation:D*_app_* = d^2^/6 t*_lag_*(4)
where d is thickness of the *stratum corneum*, which is assumed to be 10 µm. 

Moreover, the SCOE model was modified by replacing skin with dialysis membrane entrapping relevant enzymes to model skin response to the antioxidants in the presence of H_2_O_2_. The possibility to model skin response with two enzymes, namely CAT and PER, was explored by drop-casting a CAT/PER mixture (1:1, *v*/*v*, comprised of 1 mg/mL of each enzyme) onto the tip of an oxygen electrode and covered with a dialysis membrane (this setup is referred to as OE). Then, the enzyme-modified oxygen electrode (OE) was used in measurements in a similar way as SCOE. Compared to SCOE experiments, a lower H_2_O_2_ concentration, specifically 0.1 mM, was added to CBS pH 5.5, followed by the addition of 0.03 mM of Sol-CA, SLN and SLN-CA. The response of OE was evaluated for each type of nanosystem.

### 2.13. Patch Test

An in vivo irritation test was performed to evaluate the effect of SLN and ETHO applied in a single dose on the intact human skin. The occlusive patch test was conducted at the Cosmetology Center of the University of Ferrara, following the basic criteria of the protocols for the skin compatibility testing of potentially cutaneous irritant cosmetic products on human volunteers (SCCNFP/0245/99) [42,43,44]. The protocol was approved by the Ethics Committee of the University of Ferrara, Italy (study number: 170583). The test was run on 20 healthy volunteers of both sexes, under their written consent to the experimentation, in accordance with The Code of Ethics of the World Medical Association (Helsinki Declaration 1964) and its later amendments for experiments involving humans. Subjects affected by dermatitis, with history of allergic skin reaction or under anti-inflammatory drug therapy (either steroidal or non-steroidal) were excluded. Ten microliters of SLN, SLN-CA, ETHO, or ETHO-CA were posed into aluminum Finn chambers (Bracco, Milan, Italy) and applied onto the skin of the forearm or the back protected by self-sticking tape. The samples were directly applied into the Finn chamber by an insulin syringe without a needle and left in contact with the skin surface for 48 h. Skin irritant reactions (erythematous and/or edematous) were evaluated 15 min and 24 h after removing the patch and cleaning the skin from sample residual. Erythematous reactions have been sorted out into three groups, according to the erythema degree: light, clearly visible, and moderate/serious. The average irritation index was calculated as the sum of erythema and edema scores and expressed according to a scale, considering 0.5 as the threshold above which the formulation is to be classified as slightly irritant, 2.5–5.0 as moderately irritant, and 5.0–8.0 as highly irritant. 

## 3. Results

### 3.1. Preparation of Nanoparticulate Systems 

The possibility to encapsulate CA in lipid-based nanosystems was previously proposed [10,11,12]. In the present investigation, the performances of SLN and ETHO for CA administration on the skin have been considered and compared. As a preliminary step, the influence of the lipid matrix on some nanosystem physical–chemical parameters was evaluated. SLN dispersions were produced by the emulsification of molten tristearin (representing the lipid phase) with a solution of poloxamer (representing the aqueous phase) by hot homogenization followed by ultrasonication [10]. ETHO was produced by adding water into a PC ethanol solution under magnetic stirring at room temperature [11]. To encapsulate CA, in the case of SLN-CA, the drug was added to fused tristearin before emulsification, while in the case of ETHO-CA, the drug was solubilized in the PC ethanol solution before water addition. Both SLN and ETHO were milky and homogeneous dispersions, devoid of agglomerates, whose aspect was not affected by CA presence. Table 1 reports the composition of unloaded or CA loaded SLN and ETHO.

### 3.2. Characterization of Nanoparticulate Systems

#### 3.2.1. Morphological Analysis

The morphology of CA containing nanoparticulate systems was investigated by TEM and cryo-TEM, being suitable techniques to get information on the shape of colloidal systems. In the case of TEM, the stained technique allows obtaining a “negatively” colored preparation in which it is possible to view the details of the treated particles under the TEM, appearing transparent on a dark background when crossed by the electron beam. On the other hand, by cryo-TEM, a higher resolution analysis of colloidal dispersions can be obtained, using unstained vitrified specimens by the “thin film” technique [45]. This latest method enables detecting the inner structure of nanoparticles and obtaining information on their supramolecular organization. Figure 1 shows microphotographs of SLN-CA (a,b) and ETHO-CA (c,d) obtained by TEM (a,c) and cryo-TEM (b,d). The use of the different microscopic methods enabled appreciating the differences between SLN and ETHO nanostructures. Indeed, the shape of SLN-CA nanoparticles appears irregular and flat both in the case of TEM and cryo-TEM images. Instead, ETHO-CA visualized by TEM analysis showed an evanescent and fluffy aspect, which is typical of nanovesicles, confirmed by cryo-TEM, revealing the PC double layers of multilamellar vesicles with spherical and ovoidal shapes. ETHO are typically characterized by a fingerprint multilamellar pattern. This structure characterizing the membrane of ETHO makes them suitable for transdermal penetration [11].

The inner structure of SLN and ETHO has been investigated by X-ray diffraction. The SAXS profiles reported in Figure 2a confirm the lamellar organization already detected both in the case of SLN and SLN-CA [10]. From the Bragg peaks, a bilayer-to-bilayer repeat distance (i.e., the sum of the bilayer thickness plus the thickness of the water layer separating two adjacent bilayers) of 4.39 nm has been derived, independently from the presence or the absence of CA. Such a result confirms that the presence of CA did not affect the structural organization of SLN [10]. On the contrary, completely different profiles were observed in the case of ETHO. As shown in panel (b)*,* the scattering pattern for ETHO is characterized by a broad band, which is approximately centered at 1.5 nm^−1^ and typically associated to the bilayer form factor [11]. As observed by cryo-TEM, this profile confirms the presence of PC multilamellar vesicles characterized by a very disordered positional correlation between adjacent bilayers and/or by only a few stacked bilayers [25]. After the addition of CA, low-intensity Bragg peaks superposed to the bilayer form factor band become visible, endorsing the presence of PC multilamellar vesicles with an increased positional correlation between adjacent bilayers [46]. Indeed, the observed ordering of the lamellae stacking can be explained by considering that CA insertion modifies the surface charge density of the ETHO bilayers and then the balance between attractive and repulsive forces between adjacent membranes [46]. The peak position indicates a bilayer-to-bilayer repeat distance of 6.90 nm. As found by other authors, the association of phenolic compounds to vesicular systems organized in bilayers can positively influence the bilayer organization [47].

Therefore, based on cryo-TEM and SAXS analysis results, SLN are characterized by a lipid tristearin matrix organized in lamellar structures dispersed in a poloxamer aqueous phase, while the dispersed phase of ETHO is constituted of PC organized in multilamellar vesicles embedding water and ethanol. Surface charge density changes produced by CA addition are probably responsible for the observed bilayer ordering observed in ETHO-CA samples.

#### 3.2.2. Size Distribution

The size distribution of nanoparticulate systems was evaluated by PCS (Table 2). SLN and ETHO mean diameters ranged between 200 and 220 nm, which were expressed as Z average, i.e., the intensity weighted mean hydrodynamic size of nanoparticles. The dispersity indexes reported in Table 2 suggest quite homogeneous size distributions, being lower than 0.3 in the case of SLN and even smaller in the case of ETHO (Table 2) [48]. The typical intensity distributions reported in Table 2 reveal the presence of two populations, the more represented of which has a mean diameter of approximately 230 nm. CA slightly affected the mean size of the dispersed nanoparticles. The size distribution data concord with the TEM images shown in Figure 1a,c.

#### 3.2.3. Zeta Potential

Zeta potential gives an indication about the electric potential of the nanoparticulate systems, including their ionic environment [30]. Electrophoretic light scattering enabled measuring the surface charge of nanoparticles. SLN and ETHO produced in the absence of CA displayed negative and similar zeta potential values (Table 2). In the presence of CA (pKa 3.64), zeta potential values of both nanosystems became less negative, showing an absolute value reduction of 8.91 in the case of SLN-CA and 18.2 in the case of ETHO-CA. Notably, ETHO-CA are characterized by an almost neutral zeta potential value, indicating the presence of roughly the same positive and negative electron charges over the ETHO surface, which is in agreement with SAXS results, advising surface charge density changes in the case of ETHO-CA. As previously found by other authors, nanoparticles can be suitable for incorporating drugs due to electrostatic attraction [30], suggesting that CA modifies the net charge on the nanoparticle surface. The pH values of SLN and ETHO corroborated this hypothesis. Indeed, while SLN and ETHO showed pH values around 5.5, the values were lower in the presence of CA (Table 2). In this respect, the opposite charges of SLN and ETHO with respect to CA could be considered as an important factor, affecting the EE. 

#### 3.2.4. CA Encapsulation Efficiency

The EE of CA within SLN and ETHO was compared, evaluating the drug content by HPLC after nanoparticulate system ultracentrifugation. As reported in Table 2, both SLN-CA and ETHO-CA displayed very high EE values, being able to increase CA solubility with respect to water (0.5 mg/mL), up to double in the case of ETHO-CA. However, the nanoparticulate systems displayed different performances. Indeed, in the case of SLN-CA, 82% *w*/*w* of the drug weighed for preparation was associated to the disperse lipid phase, while 18% of CA was found in the dispersing aqueous phase (Table 2). Conversely, in the case of ETHO-CA, EE was total, with 100% of CA associated to the vesicles, representing the lipid disperse phase. The reduction of zeta potential value of ETHO in the presence of CA could account for the higher EE of CA with respect to SLN. Notably, the LC of the vesicles, representing the ETHO-CA disperse phase, was 7-fold higher with respect to the solid particles, i.e., the disperse phase of SLN-CA. In this regard, some consideration should be done: (a) CA is partially soluble in water (logP 1.53); therefore, it is assumed to be partly associated with the interface between the nanoparticles and the external aqueous phase and partly solubilized within poloxamer-based micelles; (b) ETHO are based on an ethanol/water mixture (30:70 *v*/*v*) where CA is slightly soluble (5.5 mg/mL) [11]; (c) the bilayer organization of PC vesicles improves CA association. In addition, the differences in the nanoparticulate production method should be considered. Indeed, in the case of SLN, the melting, ultrasonication, and cooling of tristearin could lead to the partial degradation of CA [10]. Conversely, ETHO production is devoid from heating and mechanical stresses, preserving CA from possible degradation. Thus, ETHO can be considered as more suitable for CA solubilization and encapsulation.

### 3.3. Stability of Nanoparticulate Systems

In order to compare the physical–chemical stability of SLN, ETHO, SLN-CA, and ETHO-CA, the size distribution, zeta potential, and EE of samples stored in the light at 22 °C for 6 months were evaluated. Noteworthily, all nanoparticulate systems were macroscopically stable; indeed, no phase separation or sedimentation phenomena were detected, despite the low zeta potential values evaluated by electrophoresis measurements. Variation of the mean diameters of the nanoparticulate systems as measured by PCS is reported in Figure 3. As is clearly shown, the nanoparticle and nanovesicle size remained quite stable; indeed, the SLN and SLN-CA mean size was unvaried after 6 months, while the ETHO and ETHO-CA mean diameters underwent a 10% increase after 3 months. The CA presence scarcely affected the stability of both types of nanosystems; indeed, the size profile of SLN-CA was the lowest, with a mean diameter slightly smaller with respect to SLN (15 nm), while ETHO size profiles with or without CA were completely superposable.

Regarding dispersity indexes, in all cases, the values remained quite stable after 6 months from production, being 0.27, 0.25, 0.14, and 0.22 for SLN, SLN-CA, ETHO, and ETHO-CA respectively, corroborating the size stability trend.

The zeta potential of nanoparticulate systems measured 6 months after their production decreased. Namely, the zeta potential values were −18.3 ± 0.2 and −20.4 ± 0.3 in the case of SLN and ETHO, respectively. In the presence of the drug, zeta potential became even more negative with respect to the initial values (Table 2), being −12.6 ± 0.1 and −2.5 ± 0.4 for SLN-CA and ETHO-CA. 

The chemical stability of SLN-CA and ETHO-CA was investigated, evaluating the residual CA content in nanoparticulate systems by disaggregation and HPLC analyses. Figure 4 shows the CA content variation for SLN-CA, ETHO-CA, and CA-water, which was taken as control. It is straightforward that both nanoparticulate systems were able to control CA content with respect to the simple aqueous solution; indeed, in this latest case, CA complete degradation occurred within 30 days. Notably, ETHO-CA were able to better control CA stability with respect to SLN-CA. Particularly, after 6 months of storage, CA content in SLN-CA was dramatically lower (almost 6-fold) as compared to ETHO-CA, being 13.6 and 77% for SLN-CA and ETHO-CA, respectively.

The different efficacy of nanoparticulate systems to control CA stability should be attributed to their supramolecular organization. Indeed, in the case of SLN-CA, drug protection was less efficacious with respect to ETHO-CA, since an amount of CA (18%, *w*/*w*) was not encapsulated and thus prone to degradation, being present in the dispersing aqueous phase. The encapsulated drug is supposed to be associated to the surface of SLN and thus partly exposed to degradation during storage. In addition, some authors have demonstrated that during SLN storage, some modification of the crystal lattice may occur, resulting in more stable configuration, leading to the expulsion of drug molecules [49]. This trend is corroborated by the decrease of SLN-CA zeta potential value, suggesting a less intense electrostatic attraction between CA and the surface of the SLN matrix, finally leading to a decrease of CA association. Conversely, in the case of ETHO-CA, the firm and complete association of the drug to the vesicles better preserved CA from degradation [11]. In this regard, it should be also considered that some authors have attributed to the multilamellar organization of ETHO the capability to better control drug release with respect to unilamellar vesicles [50].

### 3.4. In Vitro CA Diffusion Kinetics 

In order to compare the diffusion kinetics of CA alternatively encapsulated in SLN or in ETHO or in CA-water, Franz cells associated to nylon membranes were employed. The amount of CA diffused per unit area was plotted against the square root of time [51], yielding straight lines, as reported in Figure 5. CA diffusion from CA–water was dramatically faster with respect to SLN-CA and ETHO-CA, indicating that both nanoparticulate systems were able to control the drug diffusion. Nonetheless, the gap between CA diffusion profiles from SLN-CA and ETHO-CA highlights the different performances of these nanoparticulate systems. 

Table 3 reports F values, corresponding to the slopes of the diffusion profiles, and D values, obtained dividing F by CA concentration (mg/mL) in the different forms.

In the case of SLN-CA, CA diffusion was 7-fold slower than in the case of CA–water and 18-fold faster than the diffusion from ETHO-CA. These discrepancies evidence that in ETHO-CA, the multilamellar vesicle organization and the high ethanol concentration are able to firmly retain CA, while in the case of SLN-CA, the drug is probably partly associated on the nanoparticle surface and partly in the poloxamer dispersing phase, from which it can diffuse in a controlled fashion with respect to the simple aqueous solution, but more quickly if compared to ETHO-CA. It should be underlined that the use of a Franz cell associated to synthetic membranes cannot provide information about drug permeability through the skin. However, this model is valuable in pre-formulation studies to investigate the effect of different compositions on drug diffusion [52].

### 3.5. CA Antioxidant Activity Evaluation

The DPPH radical scavenging method enabled comparing the antioxidant activity of CA encapsulated within SLN or ETHO. The IC_50_ values of CA–water, SLN-CA, and ETHO-CA reported in Table 3 were similar, indicating that CA antioxidant capacity was maintained either under encapsulation in SLN or in ETHO. 

### 3.6. Evaluation of Skin Membrane Resistance upon Nanoparticulate System Application

To understand the interaction of each type of nanosystem with skin, skin impedance has been measured. The method allows determining the skin resistance, which is a parameter directly related to the skin barrier integrity. Indeed, a low skin resistance could indicate a damaged *stratum corneum* [33], possibly leading to a permeability increase, especially in the case hydrophilic solutes [40]. Impedance measurements of skin membrane in the presence of ETHO and SLN indicated a strong interaction with skin, as appreciable in Figure 6, showing a 5000 or 16,000 ohm increase in the case of SLN and ETHO, respectively.

The interaction of SLN and ETHO with skin was reversible, since skin resistance decreased to the same initial value upon changing SLN or ETHO back with CBS. Noteworthily, the more pronounced increase of skin resistance occurring in the case of ETHO suggests that PC vesicles established a sort of close contact and blending with skin lipids, sealing the permeability pathways for ions and hydrophilic solutes in the *stratum corneum*. 

### 3.7. In Vitro Assessment of CA Antioxidant Activity and Permeability in the Skin 

The SCOE has been used to monitor the permeability and involvement of polyphenols in antioxidant reactions, which are catalyzed by peroxidase-like enzymes present in skin. It is worth underlining that polyphenols can donate hydrogen for the reduction of H_2_O_2_ in the presence of peroxidase type enzymes available in the skin membrane [53,54]. In the SCOE model, to mimic the inflammatory condition with respect to reactive oxygen species production in the skin, H_2_O_2_ is added into the buffer solution. The H_2_O_2_ permeation through the stratum corneum generates O_2_ (Appendix A), thus increasing the electrode current due to the abundant presence of catalase enzyme in the epidermis. The possible polyphenol penetration and its involvement in the antioxidative reactions is indicated by the SCOE, showing a reduction of the electrode current. It is important to underline that many antioxidative enzymes in the skin have peroxidase-type activity (Appendix A), which in the presence of polyphenols consumes H_2_O_2_, subtracting it from the catalase reaction (Appendix A). The polyphenol CA, being a hydrogen donor, induces a peroxidase type reaction, reducing H_2_O_2_ to water. Thus, upon permeation through the skin, the electrode SCOE measures the current increase due to the O_2_ production promoted by the catalase enzyme activity in the presence of H_2_O_2_. Before starting SCOE experiments, the integrity of skin membranes has been assessed by electrical impedance measurements of their resistance. The measured values were higher than 3 kΩ, indicating the skin membrane’s integrity, i.e., intact *stratum corneum*. The plot, reported in Figure 7, indicates the relative, average amperometric current response of SCOE immersed in CBS (pH 5.5) upon the addition of Sol-CA or CA encapsulated in different nanoparticulate systems.

In Figure 7, the response set to zero indicates the steady-state current, which is usually achieved in 30 min after the SCOE immersion in CBS. Afterwards, the addition of H_2_O_2_ (0.5 mM) (marked by arrow #1) gradually increased the current response (recorded as more negative current) due to O_2_ generation by catalase reaction, confirming the H_2_O_2_ penetration into the skin. At time 0 (arrow #2), Sol-CA, SLN, SLN-CA, ETHO, or ETHO-CA were added into the measurement cell. In the case of Sol-CA and ETHO-CA, the current response (the reduction current) of the SCOE decreased, indicating the antioxidant effect of CA that hampered the H_2_O_2_ effect, which was due to peroxidase-like reactions in the skin. Noteworthily, ETHO-CA showed a more intense antioxidant activity than Sol-CA, suggesting that ETHO-CA promoted CA permeation with respect to the simple solution of the drug. This result suggests that the transdermal delivery of CA exerted by ETHO was not merely due to the ethanol presence, but instead should be attributed to the ETHO-CA supramolecular structure. Moreover, in vitro permeation parameters, such as t*_lag_* and D*_app_*, were calculated following Equation (4) and reported in Table 4. In the case of ETHO-CA, t*_lag_* was longer with respect to Sol-CA, indicating a more complicated mechanism of CA delivery. This result suggests that PC vesicles require more time to interact with *stratum corneum* lipids with respect to Sol-CA, in which case ethanol simply acts as a penetration enhancer, opening pores within *stratum corneum*.

Surprisingly, SLN-CA, SLN, and ETHO exerted an initial response in approximately 10 min, which was followed by a return to baseline values, or lower in the case of SLN-CA. This behavior in the case of SLN and ETHO could be attributed to an increased skin resistance in the presence of the nanoparticulate systems, as demonstrated by impedance measurements (Figure 6). It is known that an increased resistance of skin usually slows down the permeation of hydrophilic solutes such as H_2_O_2_ [55]. In order to better understand the amperometric response of SLN-CA, which is more intense with respect to ETHO-CA, an enzyme-modified oxygen electrode model has been employed. Namely, the OE was covered with a dialysis membrane, instead of a skin membrane, entrapping the CAT/PER enzyme system, which was employed in order to reproduce the catalase–peroxidase activity exerted within the skin. After reaching the steady-state response in the presence of H_2_O_2_, at time assigned to zero, Sol-CA, SLN-CA, ETOH:CBS (30:70 *v*/*v*), or SLN were added to the measurement cell containing the enzyme-modified OE. As shown in Figure 8, very quick and appreciable amperometric responses have been recorded in the case of Sol-CA and SLN-CA, with a higher profile and D*_app_* value in the case of Sol-CA (Table 4). The control vehicles did not give appreciable responses, as expected. The amperometric responses, quicker with respect to those obtained by SCOE model, can be attributed to the absence of the pig skin, acting as a diffusional or permeation barrier. The SLN-CA response was less intense with respect to Sol-CA (D*_app_* 2.8-fold lower), while the control vehicles responses were practically undetectable. Thus, the OE experiment evidenced the SLN-CA capability of antioxidant delivery with respect to the solution of the drug. 

The results also remarkably confirmed that the skin represented a barrier to CA permeation in the case of its administration by SLN-CA. On the contrary, when skin was present, in the case of SCOE, the ETHO-CA response, more intense with respect to Sol-CA, confirmed the permeability-enhancing effect of ETHO vesicles, promoting the CA antioxidant activity due to its passage through the skin. 

Therefore, the amperometric experiments have underlined the different performances of SLN and ETHO applied on the skin. As suggested by other authors, SLN appear less efficient to promote drug permeation as compared to ETHO. Indeed, many studies have suggested that SLN can exert an occlusive effect [56,57,58], promoting the formation of a drug depot in or on the *stratum corneum*, while other studies have demonstrated that elastic ETHO can penetrate the *stratum corneum* through the intercellular spaces and release the loaded drug [25,30,31,32]. In this respect, the transdermal effect of ETHO-CA promotes a more intense and prolonged antioxidant activity of CA with respect to SLN-CA. 

### 3.8. In Vivo Comparative Irritation Test

Although the matrix of SLN and ETHO is based on lipids generally recognized as safe, accepted by the FDA, and thoroughly compatible for cutaneous administration, it should be considered that some plant-derived compounds can cause adverse local reactions when applied on the skin [1,43]. In this regard, to in vivo evaluate the possible irritant reactions induced by the cutaneous application of nanoparticulate systems unloaded or loaded with CA, a patch test was performed on 20 health volunteers. The number of irritant reactions encountered at 15 min and at 24 h after the removal of the Finn Chamber was recorded, expressed as irritation indexes, and summarized in Table 3. The results indicate that SLN, ETHO, SLN-CA, and ETHO-CA can be classified as not irritating if applied to human skin, confirming the suitability of these nanosystems for the treatment of skin disorders [59,60].

## 4. Conclusions

This study has evidenced the differences in SLN and ETHO performances as nanoparticulate systems for CA cutaneous application. Notwithstanding the capability to encapsulate the phenolic molecule and maintain its antioxidant power, ETHO-CA can better protect CA against degradation, as compared to SLN-CA. Noteworthily, even though the in vitro Franz cell experiment suggests a more intense diffusion of CA in the case of SLN-CA and a very low diffusion in the case of ETHO, a completely different behavior was detected by the SCOE experiment based on pig skin. Indeed, this ex vivo model highlighted the transdermal potential of ETHO-CA that promoted CA antioxidant activity through the skin. The novel approach based on SCOE was effective and helpful to evaluate the transdermal antioxidant effect in the case of ETHO-CA. Furthermore, the catalase/peroxidase-modified OE enabled demonstrating the delivery of antioxidant effect by SLN-CA. Notably, the work and results obtained with skin or enzymes-modified OE reveal the limitations and the possibilities for developing novel and versatile strategies to understand the effectiveness of nanoparticulate systems as carriers for phenolic compounds.

## Figures and Tables

**Figure 1 nanomaterials-11-00171-f001:**
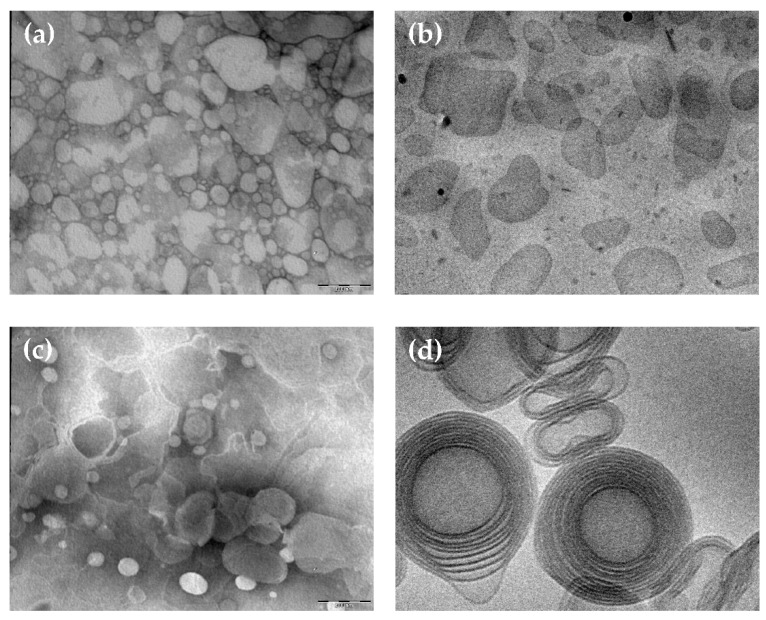
Caffeic acid containing solid lipid nanoparticles (SLN-CA) (**a**,**b**) and caffeic acid containing ethosomes (ETHO-CA) (**c**,**d**) obtained by transmission (**a**,**c**) or cryogenic transmission (**b**,**d**) electron microscopy. The bar corresponds to 150 nm in panels (**a**–**c**) and 50 nm in panel (**d**).

**Figure 2 nanomaterials-11-00171-f002:**
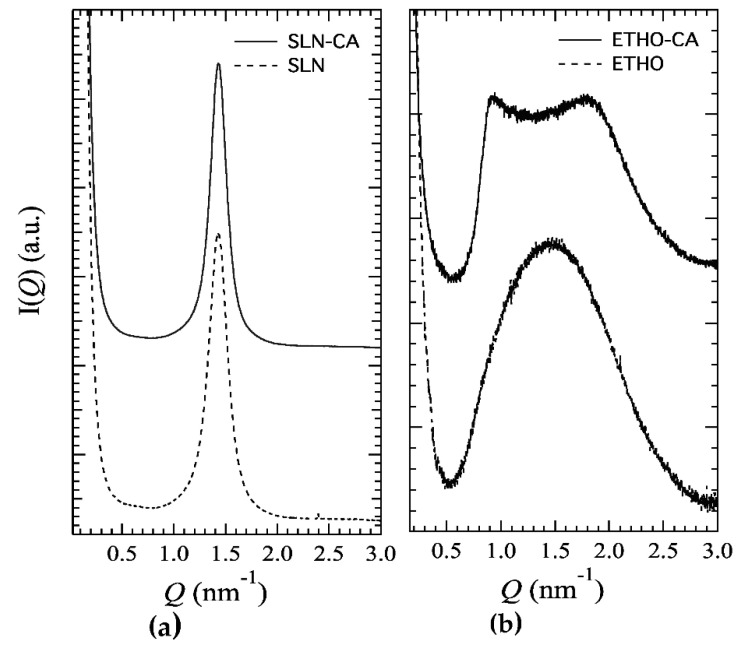
X-ray scattering profile of SLN (**a**) and ETHO (**b**) produced either in the absence (dotted lines) or in the presence (solid lines) of CA.

**Figure 3 nanomaterials-11-00171-f003:**
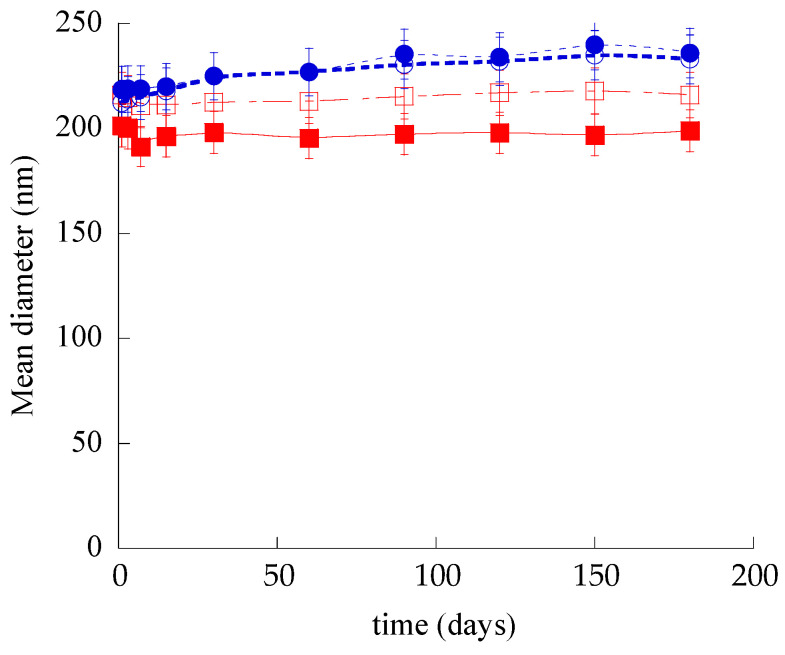
Effect of aging on SLN (open squares), SLN-CA (closed squares), ETHO (open circles), and ETHO-CA (closed circles) mean diameters measured by PCS and expressed as Z-average. Data are the mean ± s.d. on 3 different batch samples stored at 22 °C for 6 months.

**Figure 4 nanomaterials-11-00171-f004:**
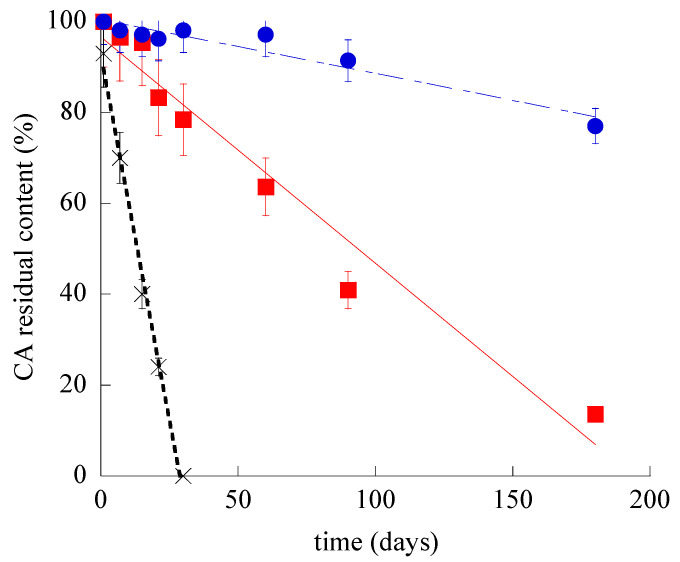
CA residual content during time for SLN-CA (squares), ETHO-CA (circles), and CA-water (crosses), as evaluated by HPLC. Data are the mean of three determinations ± s.d. on samples stored at 22 °C for 6 months.

**Figure 5 nanomaterials-11-00171-f005:**
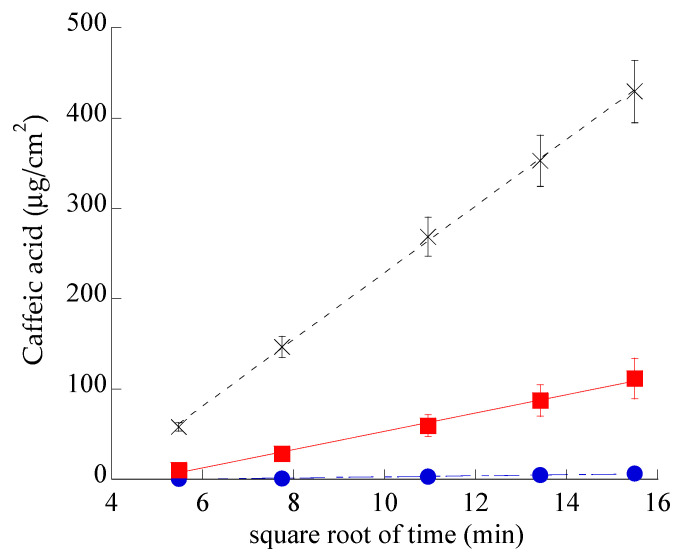
CA diffusion kinetics from CA–water (crosses), SLN-CA (squares), and ETHO-CA (circles), as determined by Franz cell experiments. Data are the mean of six independent experiments ± s.d.

**Figure 6 nanomaterials-11-00171-f006:**
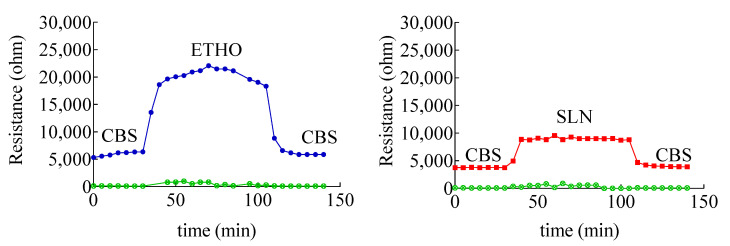
Variation of skin membrane resistance upon addition of SLN (red), ETHO (blue). The green profile refers to receiving phase solution. The skin membrane was assembled in a Franz cell equipped with four electrodes, as shown in Appendix A.

**Figure 7 nanomaterials-11-00171-f007:**
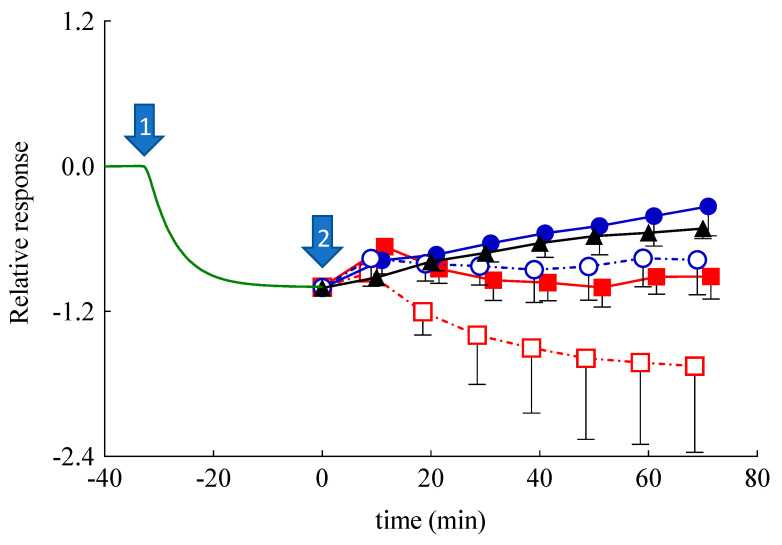
Relative, average amperometric current response of skin covered oxygen electrode (SCOE) immersed in citrate buffer saline (CBS) (pH 5.5) upon the addition of CA in different forms. Arrow #1 marks the addition of H_2_O_2_ (0.5 mM; *n* = 19), arrow #2 marks the addition of Sol-CA (CA 0.5 mM, black triangles, *n* = 3); ETHO-CA (CA 0.5 mM, blue closed circles, *n* = 6), unloaded ETHO (blue open circles, *n* = 4), SLN-CA (CA 0.5 mM, red closed squares, *n* = 3), and unloaded SLN (red open squares, *n* = 3). Time scale is adjusted by assigning t = 0 min to the moment when the formulations have been introduced to the measurement cell. The relative response is calculated, assigning a baseline current of the SCOE in CBS to zero and the current response to 0.5 mM H_2_O_2_ equal to 1.

**Figure 8 nanomaterials-11-00171-f008:**
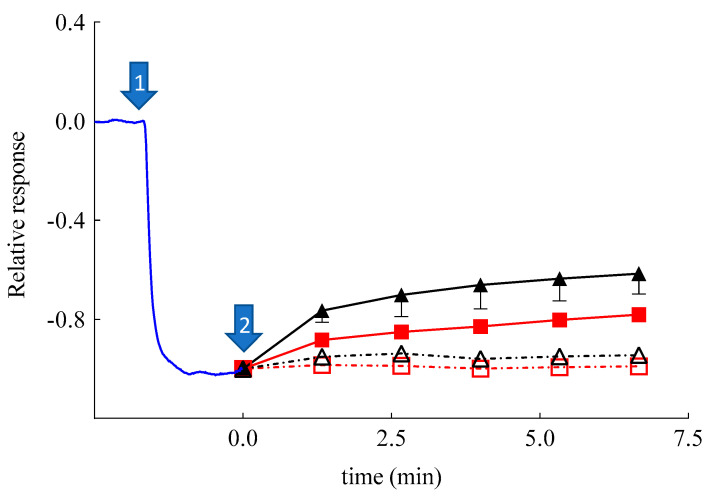
Relative, average amperometric current response on addition of H_2_O_2_ (0.1 mM) (marked by arrow #1; *n* = 13) recorded by using a catalase/peroxidase enzyme-modified oxygen electrode. Arrow #2 marks the addition of Sol-CA (CA 0.03 mM, closed triangles, *n* = 3), SLN-CA (0.03 mM CA, closed squares, *n* = 3), SLN (open squares, *n* = 3), and EtOH: CBS (30/70 *v*/*v*, open triangles, *n* = 3).

**Table 1 nanomaterials-11-00171-t001:** Composition (% *w*/*w*) of nanoparticulate systems.

Nanoparticulate System	PC ^1^	Tristearin	Poloxamer	Ethanol	Water	CA ^2^
SLN	-	5	2.38	-	92.62	-
SLN-CA	-	5	2.38	-	92.52	0.1
ETHO	0.9	-	-	70	29.1	-
ETHO-CA	0.9	-	-	69.9	29.1	0.1

^1^: phosphatidylcholine; ^2^: caffeic acid.

**Table 2 nanomaterials-11-00171-t002:** Dimensional characteristics, zeta potential, pH, and encapsulation parameters of SLN and ETHO.

Parameter	SLN	SLN-CA	ETHO	ETHO-CA
**Z-Average (nm) ± s.d.** ^1^	216 ± 12	201 ± 11	212 ± 14	219 ± 21
**Dispersity index ± s.d.** ^1^	0.28 ± 0.02	0.29 ± 0.03	0.12 ± 0.01	0.2 ± 0.02
**Intensity (nm) ± s.d.** ^2^ **Zeta potential**	230 (88%)	225 (87%)	230 (98%)	235 (90%)
	90 (12%)	75 (13%)	80 (2%)	85 (10%)
	−13.8 ± 0.1	4.9 ± 0.1	−16.2 ± 4.5	+1.99 ± 2.48
**pH**	5.46 ± 0.01	3.66 ± 0.02	5.64 ± 0.02	3.75 ± 0.01
**Encapsulation efficiency (%)** ^3^	-	82.2 ± 8.3	-	100 ± 2.0
**Loading capacity (%)** ^4^	-	1.6 ± 0.03	-	11 ± 0.10

^1^ as determined by photon correlation spectroscopy (PCS); ^2^ typical intensity distribution obtained by PCS, referring to the percentage of each population; ^3^ percentage (*w*/*w*) of drug encapsulated, with respect to the total amount used for the preparation; ^4^ percentage (*w*/*w*) of drug within nanoparticulate system, as compared to the amount of lipid used for the preparation. Data represent the mean ± s.d. of six independent experiments.

**Table 3 nanomaterials-11-00171-t003:** Fluxes, diffusion coefficients, antioxidant capacity, and irritation indexes of the indicated formulations.

Formulation Code	F ^1^ ± s.d.	CA	D ^2^ ± s.d.	DPPH S.A. ^3^	Irritation Index (Mean)	
	(μg/cm^2^/h)	(mg/mL)	(cm/h)∗10^−3^	IC_50_ (μg/mL)	0.25 h	24 h
CA–Water	36.93 ± 9.2	0.5	73.86 ± 18.4	7	-	-
SLN	-	-	-	-	0.15	0.15
SLN-CA	10.16 ± 2.5	1.0	10.16 ± 2.5	9	0.25	0.25
ETHO	-	-	-	-	0.1	0.05
ETHO-CA	0.57 ± 0.09	1.0	0.57 ± 0.09	10	0.05	0.05

^1^: Flux; ^2^: Diffusion coefficient; data are the mean of six independent Franz cell experiments; ^3^: DPPH radical scavenging activity.

**Table 4 nanomaterials-11-00171-t004:** Lag time (t*_lag_*) and apparent diffusion coefficient (D*_app_*) values of the indicated forms calculated from SCOE and enzyme-modified oxygen electrode (OE) responses.

Formulation	SCOE		Enzyme Modified OE	
	t*_lag_* (s)	D*_app_* (cm^2^/s)	t*_lag_* (s)	D*_app_* (cm^2^/s)
H_2_O_2_	500 ± 210	3.8 (±1.80) × 10^−10^	8.4 ± 1.22	0.05 (±0.0) × 10^−5^
Sol-CA	1462 ± 53	0.01 (±0.00) × 10^−10^	126 ± 18.32	7.00 (±0.1) × 10^−5^
SLN-CA	-	-	454 ± 51.61	2.50 (±4.3) × 10^−5^
ETHO-CA	5400 ± 3000	0.42 (±0.30) × 10^−10^	-	-

## Data Availability

The data presented in this study are available on request from the corresponding author. The data are not publicly available due to privacy restrictions.

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
