# Peer review of "The Potential of Caffeic Acid Lipid Nanoparticulate Systems for Skin Application: In Vitro Assays to Assess Delivery and Antioxidant Effect"

_nanomaterials, 2021, doi:10.3390/nano11010171_

Round 1

Reviewer 1 Report

Response to Authors: This manuscript very carefully characterizes the differences between solid lipid nanoparticles and ethosomes that encapsulate caffeic acid, an antioxidant that can scavenge reactive oxygen species. The characterization is very detailed involving cryo-tem, light scattering, x-ray diffraction, to characterize shape, structure, dispersity and the effect of including caffeic acid in the formulation. Encapsulation, stability, diffusion of the drug, antioxidant activity and irritation studies in humans were used to assess the performance of the delivery vehicles and their safety. This is a very complete manuscript. The authors should consider the following minor comments:

Minor Comments:

  1. In page 3, line 135 "bidistilled water was dropped to the ethanol phase..." please replace "dropped" with "added dropwise". Make a similar correction to line 138 of the same page.
  2. In page 7, line 315: Isn't the "cold method" just room temperature? Please modify the text to reflect that.
  3. The labels in Figure 1 are oddly placed. Choose inside or outside the image.
  4. Figure 1d: It appears the ethosomes have multiple lamella? Can this be controlled? Is there any correlation between loading capacity and number of lamella?
  5. Have different concentrations of CA been attempted? Since the loading capacity is low, what is the maximum amount of CA that can be loaded in the solid lipid nanoparticles and ethosomes?

Author Response

Response to Reviewer 1 Comments

This manuscript very carefully characterizes the differences between solid lipid nanoparticles and ethosomes that encapsulate caffeic acid, an antioxidant that can scavenge reactive oxygen species. The characterization is very detailed involving cryo-tem, light scattering, x-ray diffraction, to characterize shape, structure, dispersity and the effect of including caffeic acid in the formulation. Encapsulation, stability, diffusion of the drug, antioxidant activity and irritation studies in humans were used to assess the performance of the delivery vehicles and their safety. This is a very complete manuscript.

We thank the reviewer for his kind and polite comment

The authors should consider the following minor comments:

Point 1. In page 3, line 135 "bidistilled water was dropped to the ethanol phase..." please replace "dropped" with "added dropwise". Make a similar correction to line 138 of the same page.

Response 1. Lines 135 (121 in the revised manuscript) and 138 (124 in the revised manuscript) have been emended according to the reviewer suggestion.

Point 2. In page 7, line 315: Isn't the "cold method" just room temperature? Please modify the text to reflect that.

Response 2. The text has been changed to” ETHO were produced by adding water into a PC ethanol solution under magnetic stirring at room temperature [11]” line 315 (lines 302-303 in the revised manuscript)

Point 3. The labels in Figure 1 are oddly placed. Choose inside or outside the image.

Response 3. In the original manuscript (either word.docx or pdf versions) the labels were rightly positioned inside each panel. Probably a problem occurred during the electronic submission. We will kindly ask the assistant editor to pay attention to this issue in case of a possible publication.

Point 4. Figure 1d: It appears the ethosomes have multiple lamella? Can this be controlled? Is there any correlation between loading capacity and number of lamella?

Response 4. Ethosomes are typically characterized by a finger-print multilamellar pattern. This structure characterizes the membrane of ethosomes, making them suitable for transdermal penetration. Indeed, both the similarity of ethosomes multilamellar structure to that of skin stratum corneum, and the presence of ethanol, promote the passage of ethosomes through the skin. For this reason, ethosome lamellarity has not been changed. In order to clarify this concept, the phrase “ETHO are typically characterized by a finger-print multilamellar pattern. This structure characterizing the membrane of ETHO makes them suitable for transdermal penetration [11]” has been inserted at lines 326-328.

In addition, caffeic acid (CA) was probably located throughout the phosphatidylcholine bilayer, as suggested by Malekar et al. (ref #47) At this regard the authors have found that the incorporation of hydrophobic phenolics into lipid vesicular systems bilayers could thoroughly affect the bilayer organization. Thus, since the molecule is located within the bilayer of ethosomes, a multilamellar structure can better stabilize CA.

Moreover, we can assert that multilamellar ethosomes are able to better control the CA release, as suggested by other authors describing multilamellar vesicles as more suitable to control drug release with respect to unilamellar ones (Kye-Il Joo, Liang Xiao, Shuanglong Liu, Yarong Liu, Chi-Lin Lee, Peter S. Conti, Michael K. Wong, Zibo Li, Pin Wang Crosslinked Multilamellar Liposomes for Controlled Delivery of Anticancer Drugs Biomaterials. 2013 Apr; 34(12): 3098–3109. doi: 10.1016/j.biomaterials.2013.01.039). In order to explain this point, we have inserted lines 454-456: “At this regard it should be also considered that some authors have attributed to the multilamellar organization of ETHO the capability to better control drug release with respect to unilamellar vesicles [50]” and the reference 50 has been added.

Point 5. Have different concentrations of CA been attempted? Since the loading capacity is low, what is the maximum amount of CA that can be loaded in the solid lipid nanoparticles and ethosomes?

Response 5. We thank the reviewer for this observation, we checked the loading capacity values and found a typo error. Indeed, the loading capacity (LC) of SLN was 1.6 instead of 1.1 (emended in Table 2). Notwithstanding this correction, the LC value of SLN was almost 7-fold lower with respect to that of ETHO. We would like to underline that the LC refers to the amount of drug loaded with respect to the lipid phase (LP) constituting the matrix of the nanoparticulate system: LC = CA /LP × 100. It should be considered that in the case of SLN the concentration of lipid phase was 50 mg/ml, in the case of ETHO the amount of PC used was 9 mg/ml, while CA loaded was 1 mg/ml, both for SLN and ETHO. Thus, LC was low only in the case of SLN, as confirmed by EE values. Anyway, the capability of ETHO to solubilize CA has been further investigated, revealing that the upper limit was 5 mg/ml, thus 10-fold higher with respect to CA solubility in water [Hallan, et al. Design and Characterization of Ethosomes for Transdermal Delivery of Caffeic Acid. Pharmaceutics 2020, 12, 740, doi:10.3390/pharmaceutics12080740]. This result is ascribable to the CA solubility value in ethanol/water (30:70 v/v), being 5.5 mg/ml. On the contrary, in the case of SLN, 1 mg/ml was the highest amount of CA that can be loaded, indeed higher amounts led to sedimentation of the drug in the dispersion.

Reviewer 2 Report

The paper from Hallan et al. describes the development of two types of lipid based systems for the encapsulation of caffeic acid. Characterization of the systems were performed with particular emphasis on antioxidant effect studies. The work is very interesting; even so, there are some aspects that can be further improved.

Major comments:

  • The introduction is too long and focused in too well-known information. The parts that describe other works with the encapsulation of cafffeic acid are the most relevant for the work itself.
  • Regarding Figure 1, the images acquired by TEM do not provide any kind of useful information. The technique is not suitable for these samples and the too high concentration also doesn´t help the quality of the images, thereby I would withdraw figure 1a and c.
  • The authors should give an explanation for the formation of two broad Bragg peaks instead of one in the case of ETHO in the presence of CA. They only show what happens and describe that it has happened before with other phenolic compounds but do not provide any suitable explanation.
  • Regarding the size of the nanoparticles, the authors provide the mean size. However, with such polidispersity (c.a. 0.28), they must have two at least to different populations. The mean size of each population and its relative percentage should be presented. Additionally, the affirmation in lines 370-371 that they have homogenous size distributions should be withdrawn, once it is clearly wrong.
  • The stability studies should take into account not only how size and encapsulation changes with time, but also polidispersity and zeta potential.

Author Response

Response to Reviewer 2 Comments

The paper from Hallan et al. describes the development of two types of lipid based systems for the encapsulation of caffeic acid. Characterization of the systems were performed with particular emphasis on antioxidant effect studies. The work is very interesting; even so, there are some aspects that can be further improved.

We thank the reviewer for his comment

Point 1. The introduction is too long and focused in too well-known information. The parts that describe other works with the encapsulation of caffeic acid are the most relevant for the work itself.

Response 1. The introduction has been shortened in order to accomplish with the Reviewer request. Lines 42-48, 68-69, 75-77 and 91-93 have been deleted. Accordingly references 1-6 have been removed. We would like to underline that some parts of the introduction describing the differences between SLN and ETHO and their different interaction with the skin are essential for this manuscript and should not be removed.

Point 2. Regarding Figure 1, the images acquired by TEM do not provide any kind of useful information. The technique is not suitable for these samples and the too high concentration also doesn´t help the quality of the images, thereby I would withdraw figure 1a and c.

Response 2. We are sorry but we do not agree with this comment since the TEM images are useful to have an idea of nanoparticle size distribution and to corroborate PCS data. At this regard, a phrase has been inserted at lines 366-367 “The size distribution data concord with the TEM images shown in Figure 1 (a, c).”

Point 3. The authors should give an explanation for the formation of two broad Bragg peaks instead of one in the case of ETHO in the presence of CA. They only show what happens and describe that it has happened before with other phenolic compounds but do not provide any suitable explanation.

Response 3. The paragraph discussing the X-ray diffraction data from ETHO has been rewritten and now lines from 343-350 and 357-358 report an explanation for the appearance of two Bragg peaks in the case of ETHO in the presence of CA. “After the addition of CA, low intensity Bragg peaks superposed to the bilayer form factor band become visible, endorsing the presence of PC multilamellar vesicles with an increased positional correlation between adjacent bilayers [46]. Indeed, the observed ordering of the lamellae stacking can be explained by considering that CA insertion modifies the surface charge density of the ETHO bilayers and then the balance between attractive and repulsive forces between adjacent membranes [46]. The peak position indicates a bilayer-to-bilayer repeat distance of 6.90 nm. As found by other authors, the association of phenolic compounds to vesicular systems organized in bilayers can positively influence the bilayer organization [47].”; “Surface charge density changes produced by CA addition are probably responsible for the observed bilayer ordering observed in ETHO samples.”

Moreover lines 382-383 have been added “…in agreement with SAXS results advising surface charge density changes in the case of ETHO-CA”.

Point 4. Regarding the size of the nanoparticles, the authors provide the mean size. However, with such polidispersity (c.a. 0.28), they must have two at least to different populations. The mean size of each population and its relative percentage should be presented. Additionally, the affirmation in lines 370-371 that they have homogenous size distributions should be withdrawn, once it is clearly wrong.

Response 4. We thank the reviewer for his comment, however we would like to highlight that for drug delivery applications  using lipid-based carriers, a PDI of 0.3 and below indicates a homogenous population, as reported in the literature concerning the use of lipid-based carriers, such as liposome and ethosomes, (Danaei, M.; Dehghankhold, M.; Ataei, S.; Hasanzadeh Davarani, F.; Javanmard, R.; Dokhani, A.; Khorasani, S.; Mozafari, M.R. Impact of Particle Size and Polydispersity Index on the Clinical Applications of Lipidic Nanocarrier Systems. Pharmaceutics 2018, 10, 57). At this regard, the phrase: “Dispersity indexes reported in Table 2 suggest quite homogeneous size distributions, being lower than 0.3 in the case of SLN and even smaller in the case of ETHO (Table 2) [48].” has been inserted at lines 362-364, adding reference 48 as a new one. Anyway, we should apologize since in Table 2 we erroneously wrote dispersity indexes of ETHO equal to those of SLN. In the revised version of Table 2 the dispersity indexes have been corrected, being lower in the case of ETHO with respect to SLN. Moreover, to accomplish to the reviewer request, in Table 2 a typical intensity distribution has been reported for each type of nanoparticle system, with mean size of each population and its relative percentage. The phrase “The typical intensity distributions, reported in Table 2, revealed the presence of two populations, the more represented of which has a mean diameter of approximately 230 nm. CA slightly affected the mean size of the dispersed nanoparticles” has been inserted at lines 364-366.

Point 5. The stability studies should take into account not only how size and encapsulation changes with time, but also polidispersity and zeta potential.

Response 5. Dispersity indexes and zeta potential values have been measured after 6 months from nanosystem production but not reported in the original version of the manuscript. Thanks to the reviewer’s suggestion, the data have been inserted in the revised version of the manuscript at the following lines:

Lines 178-179: “In addition, zeta potential has been measured 6 months after nanoparticulate systems production.”

Lines 426-428: Regarding dispersity indexes, in all cases the values remained quite stable after 6 months from production, being 0.27, 0.25, 0.14 and 0.22 for SLN, SLN-CA, ETHO, ETHO-CA respectively, corroborating the size stability trend.”

Lines 429-432: “Zeta potential of nanoparticulate systems measured 6 months after their production decreased. Namely zeta potential values were -18.3 ± 0.2 and -20.4 ± 0.3 in the case of SLN and ETHO, respectively. In the presence of the drug, zeta potential became even more negative with respect to the initial values (Table 2), being -12.6 ± 0.1 and -2.5 ± 0.4 for SLN-CA and ETHO-CA.”

Lines 451-453: “This trend is corroborated by the decrease of SLN-CA zeta potential value, suggesting a less intense electrostatic attraction between CA and the surface of SLN matrix, finally leading to a decrease of CA association.”